# DNA Damage Repair Classifier Defines Distinct Groups in Hepatocellular Carcinoma

**DOI:** 10.3390/cancers14174282

**Published:** 2022-09-01

**Authors:** Markia A. Smith, Sarah C. Van Alsten, Andrea Walens, Jeffrey S. Damrauer, Ugwuji N. Maduekwe, Russell R. Broaddus, Michael I. Love, Melissa A. Troester, Katherine A. Hoadley

**Affiliations:** 1Department of Pathology and Laboratory Medicine, School of Medicine, University of North Carolina, Chapel Hill, NC 27599, USA; 2Department of Epidemiology, Gillings School of Global Public Health, University of North Carolina, Chapel Hill, NC 27599, USA; 3Lineberger Comprehensive Cancer Center, University of North Carolina, Chapel Hill, NC 27599, USA; 4Department of Surgery, Medical College of Wisconsin, Milwaukee, WI 53226, USA; 5Department of Genetics, University of North Carolina at Chapel Hill, Chapel Hill, NC 27599, USA; 6Department of Biostatistics, University of North Carolina at Chapel Hill, Chapel Hill, NC 27599, USA

**Keywords:** hepatocellular carcinoma, HCC, DNA repair, liver regeneration, p53

## Abstract

**Simple Summary:**

DNA repair pathways have been implicated in hepatocellular carcinoma outcomes. We found that hepatocellular carcinomas (HCC) could be separated into two groups (high and low) based on the overall expression of genes involved in DNA repair. Among the low repair group, there were three subgroups, one of which shared features of the high repair group. Given the important role of liver in metabolism and detoxification and its regenerative capacity, proliferation and DNA damage responses are critical in subdividing major biological categories of liver tumors. High repair samples showed more proliferative and regenerative signatures and had poorer outcomes versus the low repair that were more associated with the genes involved in normal liver biology. These biological groups suggest that dysregulation in endogenous liver processes promotes a pro-tumorigenic microenvironment that may facilitate tumor progression or identify tumors that require more substantial clinical intervention.

**Abstract:**

DNA repair pathways have been associated with variability in hepatocellular carcinoma (HCC) clinical outcomes, but the mechanism through which DNA repair varies as a function of liver regeneration and other HCC characteristics is poorly understood. We curated a panel of 199 genes representing 15 DNA repair pathways to identify DNA repair expression classes and evaluate their associations with liver features and clinicopathologic variables in The Cancer Genome Atlas (TCGA) HCC study. We identified two groups in HCC, defined by low or high expression across all DNA repair pathways. The low-repair group had lower grade and retained the expression of classical liver markers, whereas the high-repair group had more clinically aggressive features, increased p53 mutant-like gene expression, and high liver regenerative gene expression. These pronounced features overshadowed the variation in the low-repair subset, but when considered separately, the low-repair samples included three subgroups: L1, L2, and L3. L3 had high DNA repair expression with worse progression-free (HR 1.24, 95% CI 0.81–1.91) and overall (HR 1.63, 95% CI 0.98–2.71) survival. High-repair outcomes were also significantly worse compared with the L1 and L2 groups. HCCs vary in DNA repair expression, and a subset of tumors with high regeneration profoundly disrupts liver biology and poor prognosis.

## 1. Introduction

Hepatocellular carcinoma (HCC) is a heterogeneous type of cancer that varies vastly in clinical outcome and response to therapy. Although several studies have identified important molecular classes in HCC, uncertainty remains regarding their associations with outcomes and their complex interaction with liver regenerative processes [1,2,3,4,5,6,7,8,9,10,11,12,13]. Recently, our lab defined three distinct molecular-based subpopulations of HCC tumors, namely HCC, Blast-Like, and CCA-like, based on gene expression similarity to cholangiocarcinomas and hepatoblast cells [14]. The HCC class has prototypical HCC tumor characteristics. Blast-like has an enrichment of *TP53* mutations and HBV-positive status, exposure-related mutational signatures such as the HCC class, and transcriptional patterns similar to hepatoblasts. While the CCA-like histologically resembles HCC, molecularly, they look more like cholangiocarcinomas with similar patterns of DNA mutations (*IDH1/2*, *BAP1*), DNA damage repair mutational signatures, and transcriptional patterns. These differences in key DNA repair pathways suggest the importance of DNA repair, but an exploration of DNA-repair-specific signatures and their relationships to other cellular pathways can help elucidate their significance in HCC.

Hepatocytes, the chief functional cells of the liver, are responsible for many critical liver functions, such as detoxification, carbohydrate metabolism, lipid metabolism, and protein synthesis. If any of the regulatory pathways supporting liver function are impeded, the liver becomes more susceptible to advanced liver disease, including cirrhosis, hepatitis, and eventually, HCC [15]. These changes lead to increased rates of hepatocyte proliferation and impaired G_1_/S checkpoint [16,17]. DNA repair plays a role in normal liver function and varies as a function of the cell cycle [18]. Dysfunction in DNA repair pathways has been linked to many cancer types, but the liver’s unique capacity to regenerate adds additional complexity to HCC.

It is well-established that the liver’s regenerative capability is tightly coupled to DNA repair processes [19,20,21,22]. Upon chronic liver damage, both processes are substantially dysregulated, and there is an increased risk of genomic instability [17,23,24,25]. This dysregulation triggers a set of DNA damage response (DDR) pathways that orchestrate hepatocyte DNA repair, cell cycle arrest, and cell death. Aberrations in DNA repair and its associated pathways, such as homologous recombination (HR), mismatch repair (MMR), and nonhomologous end joining (NHEJ), have been implicated in impairing liver genomic integrity, leading to the activation of hepatocarcinogenesis and HCC development [26,27,28,29]. Given the importance of liver regeneration and DNA damage response, they could have a joint impact on prognosis and therapeutic response. Specifically, the dysregulation of DDR may determine chemoresistance, as shown in other tumor types [30,31].

Here, we investigated DNA repair defects in the TCGA HCC study using a selected panel of 199 genes representative of 15 DNA repair pathways. We sought to understand DNA repair in the context of liver homeostasis, including liver-specific gene expression, mitosis, and liver regeneration. We hypothesized that specific DNA repair patterns, along with liver regenerative capacity, could dictate prognostic groups in HCC. The further characterization of DNA repair pathways in both low- and high-repair HCC tumors, together with the integration of genomic information and clinical data, may help to better understand HCC heterogeneity in relation to outcomes.

## 2. Methods

### 2.1. Study Population and Datasets

The upper-quartile-normalized RSEM gene expression data for TCGA hepatocellular carcinoma (HCC dataset, *n* = 374) were downloaded from the GDC Legacy Archive (https://portal.gdc.cancer.gov/legacy-archive/, Last Accessed Date 2 June 2022) and log2-transformed. The data were median centered across the samples.

The mutation data, including DNA variant allele frequency (VAF), were downloaded from Ellrott et al. [32] (https://gdc.cancer.gov/about-data/publications/mc3-2017, Last Accessed Date 7 June 2022). The processed RNA data were retrieved from TCGA PanCancer (https://gdc.cancer.gov/about-data/publications/pancanatlas, Last Accessed Date 7 June 2022) [33]. For each somatic mutation in the VAF file, we calculated RNA fragments’ allele counts.

### 2.2. Classification of DNA Repair Groups in RNA Expression Data

We curated a list of 199 genes representing the regulators of many DNA repair pathways (Appendix A). The panel comprised the following DNA repair pathways: mismatch repair (MMR), nucleotide excision repair (NER), translesion synthesis (TLS), Fanconi anemia (FA), base excision repair (BER), nucleotide metabolism, template switch, poly ADP-ribose polymerases (PARP), checkpoint, DNA replication factors/cell cycle, homologous recombination (HR), nonhomologous end joining (NHEJ), alternative end joining (Alt-EJ), cancer-testis antigens (CTAs) (including *HORMAD1* and *MAGEA4*, which are pathological cancer-specific activators of HR and TLS, respectively [34,35]), and the APOBEC cytosine deaminase family.

The 199-Repair score was calculated as the median of the DNA repair genes. We applied the R package mclust (v5.4.7) [36] to select the optimal cut point, and based on the determined cut-point value of 0.1, the samples were classified as low repair or high repair. The hepatitis B virus (HBV) was detected in HCC tumor RNAseq data via VirDetect [37]. The expression data were visualized using ComplexHeatmap (v2.8.0) [38] in R. All analyses were performed using the R Statistical Software (v4.1.0, R Core Team 2021, Vienna, Austria) [39] unless otherwise noted.

### 2.3. Define Mitotic and Regeneration Patterns in HCC

We calculated the mitotic index score using the median of 10 mitotic pathway genes—*BUB1*, *BUB1B*, *BUB3*, *CDC20*, *CDH1*, *ESPL1*, *MAD1L1*, *MAD2L1*, *PTTG1*, *TRIM69*—from the 199 DNA repair gene panel. A regeneration score from Colak et al. [40] was derived from their regeneration activation and inhibition gene sets in by multiplying the activation gene expression values by +1 and the inhibition gene expression values by −1, and after combining them, the median value was calculated. We compared the continuous mitotic and regeneration score components of DNA repair groups using the Wilcoxon signed-ranked tests.

### 2.4. Interrogation of the Biological States and Processes in DNA Repair Groups

Gene set enrichment analysis (GSEA) [41,42] was performed by comparing the high-repair group vs. the low-repair group for all HCC tumors, and significance was determined by using a false-discovery rate (FDR) <0.25. We utilized the MSigDB Hallmark gene set and excluded any genes overlapping (103 genes) with our DNA repair gene panel to control for the fact that the groups were derived from the RNA expression data. Normalized enrichment scores (NES) were plotted for Hallmark pathways identified as significant for the high-repair and low-repair groups. We calculated pathway scores for selected significant pathways. Hallmark pathway scores and liver-specific markers were compared, stratified by the repair groups, with tumor-adjacent normal tissues using pairwise *t*-tests to investigate signatures of liver’s biological function.

### 2.5. Association of p53 and HRD

To identify mutations associated with the DNA repair groups, we used a Fisher’s exact test between low and repair groups restricting analyses to genes mutated in at least 5% of the study. We also included *ALB* gene as it is involved in liver metabolic pathways and mutated in HCC. Genes in HRD pathway with mutations were selected based on Chen et al. [43]. We used a previously validated RNA signature that aggregates the information on TP53-dependent genes to classify the *TP53* functional status (mutant-like or wild-type) based on a similarity-to-centroid approach. The *TP53* mutational status was classified based on the presence (*TP53* mut) or absence (*TP53* WT) of somatic *TP53* mutations. We compared the *TP53* RNA variant allele frequency (VAF) vs. DNA VAF stratified by repair groups using a linear model. Homologous recombination deficiency (HRD) scores, including the loss of heterozygosity (LOH) [44], large-scale transitions (LST) [45], and the number of subchromosomal regions with allelic imbalance extending to the telomere (NtAI) [46], were extracted from supplemental data in Knijnenburg et al. [47]. The scores were dichotomized at a cut point of 20 based on the distribution of HRD scores in the liver to ensure the HRD status is tissue-specific similar to prior studies in the literature [48,49,50].

We estimated the relative frequency differences (RFDs), representative of the difference between an index group and a reference group in the proportion of individuals exhibiting a given clinical or demographic feature, between the high-repair/regenerative (index) group and the low-repair (referent) group. RFDs and 95% confidence intervals (CIs) were estimated using generalized linear models with binomial distributions and identity link functions.

### 2.6. Low-Repair Tumor Classification

To unmask the differences in the low-repair group, we removed the samples classified in the high-repair/regenerative group from the TCGA HCC expression data. We performed a clustering analysis using the R package ConsensusClusterPlus (v1.56.0) [51] on the low-repair group to evaluate the expression differences present in the low-repair group.

### 2.7. Clinical Variables and Survival Analysis

We extracted the TCGA clinicopathologic and survival data from Liu et.al [52]. The clinical data were filtered to samples with ≥94% of available clinical annotations across the study for race, gender, age, pathologic stage, pathologic primary tumor (pT), grade, and survival. The AJCC TNM Classification 2010 (7th Edition) [53] was used for pathologic primary tumor (pT) and the samples classified by the AJCC TNM Classification 6th Edition [54] were converted to those classified by the 7th edition. The samples classified based on the AJCC 4th and 5th edition TNM staging system [55,56] were excluded from our analyses due to the lack of sufficient information for conversion to the 7th edition. Kaplan–Meier curves with log-rank tests were generated using the R package survminer (v0.4.9) [57]. Univariate Cox regression was used to determine the significance of DNA repair groups, sex, race, molecular subtype, stage, grade, and HBV status. The models were stratified by pathological tumor stage and tumor grade to determine whether associations between the repair status and survival held. Continuous variable comparisons were made across the repair groups using the Wilcoxon signed-rank test or ANOVA as indicated. Statistical analyses were performed using R unless otherwise noted.

## 3. Results

### 3.1. HCC Tumors Exhibit Two Groups Based on Expression of 199 DNA Repair Genes

To characterize DNA repair patterns in TCGA hepatocellular carcinoma (HCC) samples, we calculated an RNA-based DNA repair score based on 199 DNA repair genes and identified an optimal cut point to distinguish the groups. HCC tumors were classified into two groups based on the DNA repair gene expression: low-repair (*n* = 216) and high-repair (*n* = 158) (Figure 1A, Appendix A). The high-repair group includes many upregulated genes across all the DNA repair pathways, pointing to the high activity of DNA repair genes. The TCGA HCC samples classified as high-repair were significantly associated with Blast-like and CCA-like molecular subtypes, Asian race, the enrichment of HBV-positive cases, and higher tumor stage, primary tumor (pT), and grade (Table 1). The median age at diagnosis was significantly lower in the high-repair group. There were no significant differences in vascular invasion between the repair groups. Low-repair samples represent prototypical HCC tumors, as most were classified in the HCC molecular subtype and were significantly enriched for lower-stage and -grade tumors.

Based on the overall high DNA repair expression seen in the high-repair group, we hypothesized that this group may have a high activity of mitotic and liver regenerative pathways. To investigate, we examined the mitotic and liver regenerative pathway patterns stratified by the repair groups. The mitotic index score and regenerative score were significantly increased in the high-repair group compared with the low-repair group (*p* < 0.001, Figure 1B,C) (Appendix A). 

To further interrogate the biological processes occurring in these groups, we performed gene set enrichment analysis (GSEA) using the Hallmark gene signature set (excluding the genes overlapping with our DNA repair gene set) to identify the differential pathway signatures between the high- and low-repair groups (Appendix A). The high-repair group was significantly enriched for five pathways that are characterized by cell cycle gene sets (G2M checkpoint, E2F targets, mitotic spindle), spermatogenesis, and MYC targets. By contrast, the low-repair group had eight significantly enriched pathways, all the gene sets related to liver biology and function (adipogenesis, xenobiotic metabolism, fatty acid metabolism, coagulation, bile acid metabolism, oxidative phosphorylation, peroxisome, and reactive oxygen species). We further looked at the signature scores for pathways that were of substantive interest due to their significance in liver biology and DNA repair mechanisms. The adipogenesis and fatty acid metabolism signature scores were significantly higher in the low-repair group but still not as high as the tumor-adjacent normal tissue (*p* < 0.001, Appendix A). MYC target genes were significantly upregulated in the high-repair samples compared with the low-repair group (*p* < 0.001 for all pairwise comparisons, Appendix A). The high-repair group displayed significantly higher expression of cell cycle and hepatoblast marker *AFP* than the low-repair group and tumor-adjacent normal tissues (*p* < 0.001 for all pairwise comparisons, Appendix A). The low-repair group also had significantly increased expression levels of liver markers *ALB* and cytochrome P450 (*CYP450*), but the expression levels were still not as high as those of the normal liver (*p* < 0.01, Appendix A). The low-repair group displayed a more preserved liver biology gene expression, while the high-repair group showed decreased liver biological gene expression and increased DNA repair dysregulation.

### 3.2. High-Repair Classes Are Associated with p53 Functional Status and TP53 Mutation Status

Given that ~36% of HCC tumors have *TP53* mutations and considering the mutual regulation between cell cycle and p53, we hypothesized that p53 plays a vital role in liver repair dysfunction and genomic instability. We assessed the mutational landscape based on the repair groups. Of the 781 recurrently mutated genes present in at least 5% of the liver samples, only *TP53* and *CTNNB1* were significantly differently mutated between the repair groups. The high-repair group had a significantly higher rate of *TP53* mutations (44.9%) than the low-repair group (19%, *p* < 0.001, Table 2). Interestingly, *CTNNB1* mutations were significantly enriched in the low-repair group (32.4%) compared with those enriched in the high-repair group (18.4%) (*p* < 0.001). *CTNNB1* mutations frequently occur in HCCs, leading to the activation of the WNT/β-catenin signaling pathway in 30–50% of HCC cases [58]. Another key liver gene, *ALB,* was more frequently mutated in the low-repair group (19.4%) than in the high-repair group (11.4%), though it did not reach significance. The key HRD genes (*ATM*, *POLE*, *BRCA1/2*, *BARD1*, *BRIP1*) exhibited less than a 5% mutation rate in both groups, except for the DNA damage response pathway gene *AXIN1*, which was higher in the high-repair group (9.5%) than the low-repair one (6.0%).

To further consider the role of master regulator *TP53*, we used a previously validated RNA signature to classify tumors for the p53 functional status (mutant-like/Wild-Type [WT]) [59]. The patterns of expression for the p53 gene signature are shown across the HCC tumors in Figure 2A. Two groups are evident: one enriched for p53 mutant-like and the other enriched for p53 wild-type (WT). The high-repair group was significantly enriched for the tumors classified as p53 mutant-like, accounting for 75% of the p53 mutant-like HCC samples (*p* < 0.001, Appendix A).

We evaluated whether the p53 status (RNA), *TP53* mutational status (DNA), HRD, and HBV status were associated with the DNA repair groups. The high-repair group had enrichment for the p53 mutant-like status (RNA), *TP53* mutation status (DNA), HRD-high status, and HBV-positive status (Figure 2B, Appendix A). In the high-repair group, the mutant p53 signature (RFD: 66.5%, 95% CI 58.5–73.5) was even more prevalent than the *TP53* DNA mutation status (RFD: 26.0%, 95% CI 16.6–35.2) (Figure 2B). Increased HBV-positive status tracks with the correlation of increased *TP53* mutations in HBV+ HCC tumors. The high-repair tumors had significantly higher p53 score and HRD scores than the low-repair tumors (*p* < 0.001) (Appendix A). The HRD-high tumors were unequally distributed between the low-repair and high-repair tumors (RFD: 34%, 95% CI 24.2–44.0) (Figure 2B). The high-repair samples are more likely to be p53 mutant-like and have a high DNA variant allele frequency (VAF), indicating mutation and single copy loss or loss of heterozygosity, in addition to having a high RNA expression of the mutant allele (Appendix A). These results imply that there was strong *TP53* dysfunction in the high-repair group. The differences observed in the p53 mutant status underscored the importance of p53 in liver tissue homeostasis.

### 3.3. Clustering Analysis, Which Reveals Three Subgroups within Low-Repair Groups

The overall, generally high expression of DNA repair genes in the high-repair/regenerative group may mask heterogeneity within the low-repair group. To assess this, we performed a consensus clustering analysis in only the samples classified as low repair. The clustering analysis revealed three subgroups—L1, L2, and L3—with increased heterogeneity across the DNA repair genes (Figure 3A). L1 was significantly decreased for DNA repair activity compared to L2 and L3, but L3 showed greater similarities to the higher repair activity seen in the high-repair group (Figure 3B, L1 vs. rest *p* < 0.001). L1 and L3 had higher expression of homologous recombination (HR)-related genes (*BRIP1, BRCA2, BARD1, RAD51AP1*). L2 and L3 were characterized by higher expression levels of cell cycle and mitotic checkpoint genes (*BUB1* and *BUB1B*). L2 showed higher expression levels of replication factors (*POLD1, RB1,* and *HORMAD1)*. When we examined the associations between clinicopathological features and the risks between the subgroups, only race, grade, and HBV status were significant (Table 3). L1 was associated with lower-grade tumors, while L2 and L3 were significantly enriched for higher-grade tumors (*p* < 0.001). Asian race and HBV-positive status were more associated with L2 and L3 than with L1 (*p* = 0.002 (*p*-value based on Asian race vs. White), *p* = 0.001). There were no associations with pathologic tumor stage, pathologic primary tumor stage (pT), or vascular invasion across the low-repair groups. We assessed the differences in the *TP53* functional status, *TP53* mutational status, and HRD status in the low-repair subgroups (Appendix A). L2 and L3 were significantly enriched for the p53 mutant-like (57.8% and 42.2%, respectively) and *TP53* mutations (29.3% and 61.0%, respectively), which were mostly lacking in L1 (p53 mutant-like 0%, *TP53* mutation 9.7%, *p* < 0.001). Additionally, L2 and L3 had significantly higher HRD scores than L1 (*p* < 0.001). We also explored immune cell specific signatures and did not find any strong associations with DNA repair classes.

### 3.4. High-Repair Group Has Worse Overall Survival and Progression-Free Survival

We investigated the survival differences between the low-repair subgroups and the high-repair group. While there were molecular differences in the samples classified as L1 and L2, these features did not affect the clinical survival rate, so we combined L1+L2 samples for the survival analysis. The progression-free and overall survival rates censored at five years were compared across the low- and high-repair groups (Figure 4) (Appendix A). Using the Cox proportional hazards model, high-repair samples revealed significantly worse progression-free (hazard ratio (HR) 1.78, 95% CI 1.26–2.49) and overall (HR 1.97, 95% CI 1.29–2.99) survival than the L1+L2 tumors. By contrast, the L3 group showed an intermediary level of survival that was significantly worse than the progression-free (HR 1.24, 95% CI 0.81–1.91) and overall (HR 1.63, 95% CI 0.98–2.71) survival outcomes observed in the L1+L2 tumors but better than the survival rates of high-repair tumors. When we added the pathologic tumor stage and grade to the model with our repair groups, both L3 and high-repair tumors retained their significance for worse overall survival but only the high-repair tumors retained their significance for worse progression-free survival compared with the referent L1 + L2 tumors. Our DNA repair score classification schema identified a set of patients with worse outcomes based on intermediate and high DNA repair gene expression.

## 4. Discussion

We used a curated DNA repair panel to identify DNA repair expression classes in TCGA HCC tumors. We identified two separate DNA repair groups (low-repair and high-repair/regenerative) with distinct biological patterns. The low-repair group was characterized by classical HCC tumor features and lower grades, while the high-repair group had very high DNA repair activity, and was associated with Blast-like and CCA-like molecular subtypes, stage, Asian race, and worse prognosis. The high-repair group overshadowed subtleties in the low-repair group, and therefore, clustering solely in the low-repair group elucidated three subgroups: L1, L2, and L3. The L1 and L2 subgroups were characterized by liver pathways and lacked regenerative processes associated with the high-repair group. L3 retained the expression of liver-related genes but was more similar to the DNA repair expression and survival outcomes of the high-repair group. Based on these findings, L3 tumors may represent a transitory/transdifferentiation HCC group that may become high repair over time.

Previous studies have detected similar molecular classes. Most studies emphasized two main groups based on the DNA repair genes’ RNA expression [8,12,13,60]. Three previous studies identified two DNA repair groups: Oshi et al. found low- and high-repair groups based on a gene set that included 150 genes [12]; Lin et al. classified DNA-repair-activated and -suppressed groups using 276 genes [13]; and Chen et al. found low- and high-risk DNA repair groups based on 23 DDR-related gene pairs [60]. Across these studies, similar to our results, the high-repair-activated samples were associated with worse survival, and these authors further found that high-repair-/DDR-activated samples were associated with distinct immune profiles, poor differentiation, elevated intratumor heterogeneity, and mutation burden. Of these three studies, gene overlap with our 199-gene set varied (overlap was 26, 122, and 11 genes for Oshi et al., Lin et al., and Chen et al., respectively). The similarities observed across these studies, despite the different number of overlapping genes, underscores that these high-repair patterns are robust and can be detected with few features. One advantage of our gene list, which covers many DNA repair pathways, was that we included several essential DNA repair genes such as *BUB1, BUB1B, RAD51AP1, RB1*, and *HORMAD1* that are known to have significant clinical relevance [61,62,63,64,65,66] and helped us to define the heterogeneity among the low DNA repair group. The mitotic and replicative factors, along with HR expression, helped to distinguish low-repair groups. While there is agreement among groups on the two-classification schema, extending beyond two groups has proven more difficult with little resolution. For example, one lab found four groups using a combination of manual and cluster-based methods. In comparison, we used a data-driven clustering approach, and while both studies found four groups, there were different gene sets utilized, and the groups considerably varied. It is clear, regardless of the size of the gene set, that we can identify high-repair groups, but more resolution is needed to identify low-repair samples.

Liver regeneration is a key liver feature that plays a role in HCC development and suggested to support a tumorigenic environment [67]. Our results suggest that liver regeneration includes the chronic activation of the DNA damage response pathway [68]. High-repair/regenerative tumor phenotypes were related to high-grade and Blast-like molecular subtypes. Similarly, samples classified within the high-repair group showed increased wound healing (i.e., increased mitotic activity), increased regenerative activity, and increased p53 dysfunction, indicating that these processes act in tandem. Many of these pathways have been implicated in micrometastases and de novo cancers following surgical resection for liver diseases and HCC [67,69,70,71], emphasizing that these pathways are interconnected and necessary for liver homeostasis. 

We further contextualized DNA repair in association with other established pathways implicated in HCC (p53, HBV status, and liver inflammation) to reveal the underlying dysregulation. Prevalent p53 dysfunction was a hallmark of the high-repair group. In particular, there was a great difference in RFDs between the RNA-based p53 status (66.5%) vs. *TP53* mutations (26.0%). Integration with RNA-based measures provides key insights because mutations alone do not cover the full molecular portrait, and in the setting of low-burden tumors such as in HCC, these methods allow us to still investigate pathway dysregulation. Cross-platform analysis of the p53 pathway revealed the importance of p53 regulation in HCC heterogeneity. Due to the role of p53 in mediating senescence after liver injury, it is responsible for the regulation of fibrosis and may prevent deterioration leading to HCC [72]. In addition, the mitotic activity is tightly and negatively controlled by p53 [73], and mitotic dysregulation is strongly associated with improper cell division and aneuploidy [74] and, therefore, may promote inflammatory environments [75,76,77].

With respect to HBV, TCGA includes representation from Southeast Asian patients where a major risk factor is HBV infection. HBV subverts the DNA damage repair (DDR) pathways, creating an inflammatory environment that promotes hepatocarcinogenesis and HCC development. Data have revealed a link between chronic HBV infection and immune-induced liver injury [78,79,80]. This injury can result in hyperactive DDR and, in turn, lead to a more aggressive chemoresistant unresectable liver cancer. Based on our findings, HBV-positive HCC tumors are associated with increased expression levels of cell cycle and mitotic genes. We also observed varying expression levels of DNA repair genes in HBV-negative tumors. While the previous discourse has highlighted viral hepatitis, our results suggest there is greater heterogeneity in HCC than previously appreciated, and outcomes may not be solely driven by the HBV infection status. This emphasizes the importance of expanding beyond the comparative paradigm of viral hepatitis vs. nonviral classification and evaluating other significant risk factors such as nonalcoholic fatty liver disease (NAFLD) and nonalcoholic steatohepatitis (NASH) in HCC clinical outcomes.

The major advance of our work was in showing that separating high-repair/regenerative tumors allowed the improved characterization of heterogeneity among low-repair HCCs with more intact liver signaling and homeostasis. This helped identify the L3 tumor group, which was found to have some preserved liver biology and functional status but higher HRD and DNA repair dysfunction. L3 also had a higher frequency of p53 mutant-like and *TP53* mutations. We also noted differences in the racial profile, grade, HBV status, and molecular differences in L1 and L2, but these features did not dictate the differences in survival outcomes between the samples. By stratifying based on the repair status, we identified novel classes that predict outcomes. Regardless of the tumor grade and stage, L3 and high-repair groups had worse prognoses than the L1 and L2 tumors, further underscoring the HCC heterogeneity and the interaction of multiple factors that lead to the differences in outcomes. All the study participants had localized disease with very few final pathologic metastases, making them eligible for surgical intervention. TCGA has a short length of follow-ups and lacks liver disease scoring (Child Pugh classification, Ishak fibrosis stage) and treatment data, limiting our ability to investigate the role of our DNA repair classifiers in the extent of liver damage and HCC patient therapy. DNA damage response dysregulation is often enhanced in HCC, resulting in poor response to DNA-damaging anticancer therapies [25]. Researchers observed that high-repair HCC tumors treated with conventional chemotherapy resulted in worse overall survival than high-repair-status patients who received immune checkpoint inhibitors [81]. The assessment of DNA damage and repair dysregulation may pose another method of stratifying patients based on DNA repair status for improved treatment management and selection. More comprehensive and standardized annotation for clinical data elements will be essential to better understand the associations between DNA repair defects, liver regenerative mechanisms, and risk factors in HCC.

## 5. Conclusions

In summary, this work expands on the findings of prior studies to identify four DNA repair classes in HCC (L1, L2, L3, and high-repair/regenerative) that are associated with distinct clinical prognoses. Our data suggest that DNA repair and liver contribute to HCC heterogeneity, resulting in variable clinical outcomes. Our signature suggests potential prognostic groups in HCC with distinct biology and plausible roles in clinical utility. Future work should evaluate heterogeneity in DNA repair in association with specific chemotherapeutic regimens to address the critical need for improved treatment strategies for this poor-prognosis cancer type.

## Figures and Tables

**Figure 1 cancers-14-04282-f001:**
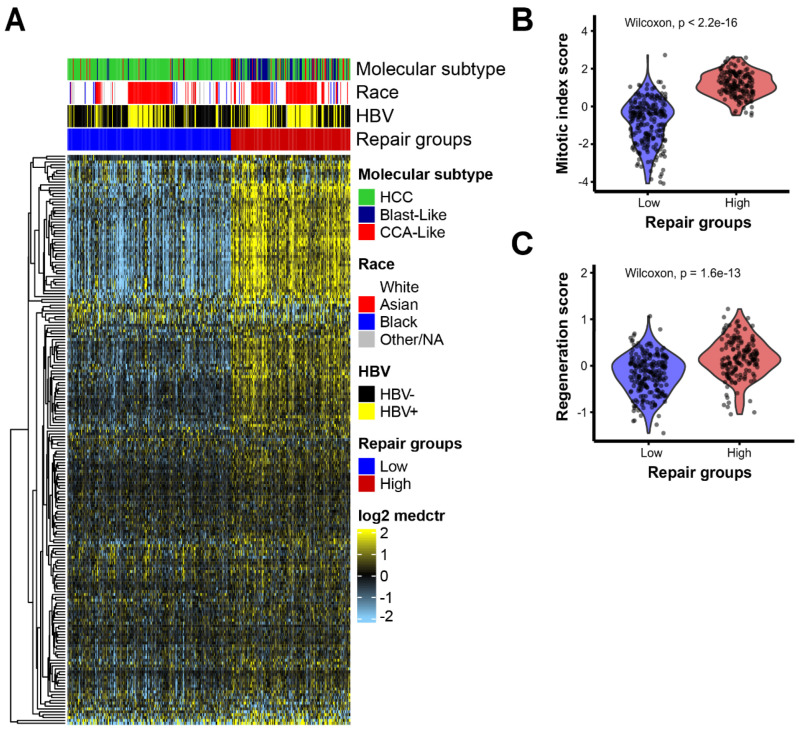
HCC tumors separated into two distinct groups based on DNA repair pathway gene expression: (**A**) expression heatmap of 199 DNA repair genes ordered by repair groups. The heatmap scale is from low expression (light blue) to high expression (yellow). Rows represent DNA repair genes and column samples. Annotation tracks show molecular subtype, race, HBV status, and Repair groups. Violin plots of (**B**) mitotic index score and (**C**) regeneration score relative to repair groups, low repair (blue) and high repair (red); *p*-value based on Wilcoxon test between groups.

**Figure 2 cancers-14-04282-f002:**
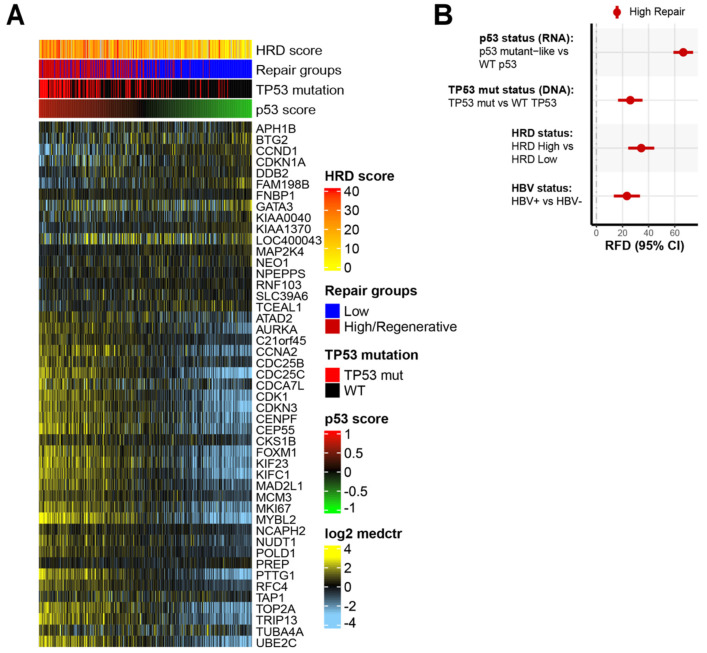
High-repair classes are associated with dysfunctional p53 functional status and TP53 mutation status: (**A**) expression patterns of p53 gene signature with samples (columns) ordered by p53 score and genes (rows) sorted by p53 mutant-like genes. The p53 score is scaled low (green) to high (red). The heatmap scale is low expression (light blue) to high expression (yellow). Annotation tracks show HRD score, DNA repair groups, *TP53* mutation status, and p53 score; (**B**) relative frequency difference (RFD) analysis of features in high-repair group compared with low-repair group. Features include p53 mutant status, TP53 mutation status, HRD status, and HBV status. The 95% confidence intervals (CIs) are included for each measure.

**Figure 3 cancers-14-04282-f003:**
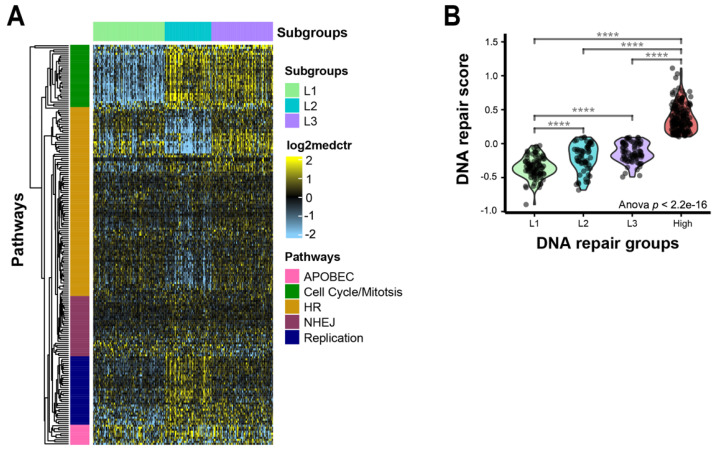
DNA repair pathway gene expression heterogeneity within the low-repair group: (**A**) heatmap DNA repair genes in low-repair subgroups. The heatmap’s scale is low expression (light blue) to high expression (yellow). Rows are DNA repair genes, and columns are samples. Annotation track shows clusters: L1 (light green), L2 (turquoise), and L3 (purple). Major gene pathways in each gene cluster are annotated on the rows: APOBEC (pink), cell cycle/mitosis (green), HR (gold), NHEJ (plum), and replication (navy); (**B**) repair gene score identifies distinct features across early and advanced HCC tumors. Violin plot of repair score by repair groups; **** indicates *p* < 0.0001 for two-sample t-test between high-expression group and L1.

**Figure 4 cancers-14-04282-f004:**
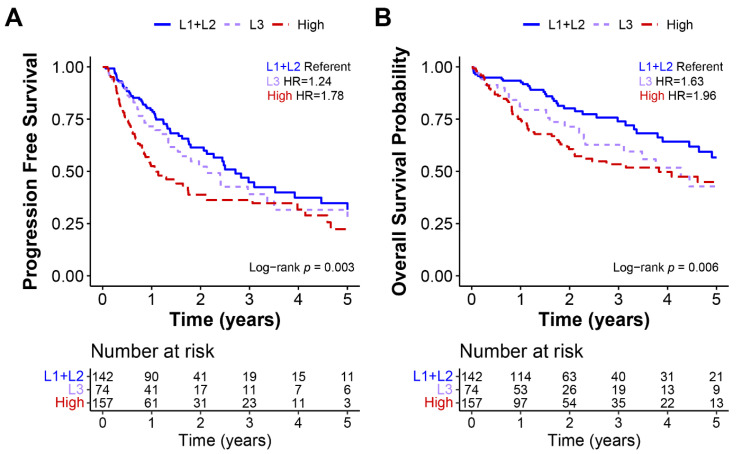
Low-repair subgroups L1 and L2 had better progression-free and overall survival. Kaplan–Meier curves of TCGA HCC data for (**A**) progression-free and (**B**) overall survival. All survival data were censored at 5 years, and hazard ratios, 95% CIs, and log-rank *p*-values were calculated for each measure.

**Table 1 cancers-14-04282-t001:** Patient characteristics of TCGA HCC study, overall and stratified by DNA repair groups. Column values in percentages.

	Overall	Low	High	*p*-Value	RFD [95% CI]
*n*	374	216	158		
Age at diagnosis					
Mean ± SD	59.48 (13.47)	60.74 (± 13.89)	57.75 (± 12.71)	0.034	-
Sex					
Female	121 (32.4)	65 (30.1)	56 (35.4)	0.327	REF
Male	253 (67.6)	151 (69.9)	102 (64.6)		−5.35 [−15.0–4.3]
Race ^a^					
White	185 (51.1)	117 (57.1)	68 (43.3)	0.025	REF
Asian	160 (44.1)	78 (38.0)	82 (52.2)		14.2 [3.92–24.4]
Black	17 (4.7)	10 (4.9)	7 (4.5)		−0.42 [−4.79–3.95]
Molecular Subtype					
HCC	275 (73.5)	198 (91.7)	77 (48.7)	<0.001	REF
Blast-Like	66 (17.6)	9 (4.2)	57 (36.1)		31.9 [24.0–39.9]
CCA-Like	33 (8.8)	9 (4.2)	24 (15.2)		11.0 [4.82–17.2]
AJCC Pathologic Tumor Stage ^b^					
I	173 (49.4)	116 (57.7)	57 (38.3)	0.001	REF
II	87 (24.9)	44 (21.9)	43 (28.9)		6.97 [−2.28–16.2]
III/IV	90 (25.7)	41 (20.4)	49 (32.9)		12.5 [3.11–21.9]
AJCC Pathologic Primary Tumor (pT) ^c^					
T1	178 (51.4)	120 (61.9)	58 (38.2)	<0.001	REF
T2	88 (25.4)	40 (20.6)	48 (31.6)		11.0 [1.68–20.3]
T3/T4	80 (23.1)	34 (17.5)	46 (30.3)		12.7 [3.74–21.9]
Grade ^d^					
G1/G2	233 (62.3)	155 (73.1)	78 (49.7)	<0.001	REF
G3/G4	136 (36.4)	57 (26.9)	79 (50.3)		23.4 [13.6–33.3]
Cirrhosis/Fibrosis ^e^					
None	75 (34.9)	49 (36.6)	26 (32.1)	0.604	REF
Cirrhosis/Fibrosis	140 (65.1)	85 (63.4)	55 (67.9)		4.47 [−8.56–17.5]
HBV infection ^f^					
Negative	217 (59.0)	146 (68.9)	71 (45.5)	<0.001	REF
Positive	151 (41.0)	66 (31.1)	85 (54.5)		23.4 [13.4–33.4]
Vascular invasion ^g^					
None	208 (65.4)	115 (65.0)	93 (66.0)	0.948	REF
Micro/Macro	110 (34.6)	62 (35.0)	48 (34.0)		0.98 [−9.57–11.4]

^a^ Excludes samples noted as other and NA (*n* = 12). ^b^ Excludes samples without pathologic stage annotation (*n* = 24). ^c^ Excludes samples based on AJCC 4th and 5th editions and noted as NA (*n* = 28). ^d^ Excludes samples without grade annotation (*n* = 5). ^e^ Excludes samples without annotation (*n* = 159). ^f^ HBV infection determined by >5 HBV reads from RNA–seq. ^g^ Excludes samples without annotation (*n* = 56).

**Table 2 cancers-14-04282-t002:** Overall mutation rate for frequently mutated, liver metabolic, and DNA repair genes in HCC stratified by DNA Repair groups.

Frequently Mutated Genes			
Gene	Low Repair (*n* = 216)	High Repair (*n* = 158)	
TP53	41 (19.0%)	71 (44.9%)	***
CTNNB1	70 (32.4%)	29 (18.4%)	***
Liver-metabolic-mutated genes			
ALB	42 (19.4%)	18 (11.4%)	ns
HRD-mutated genes			
AXIN1	13 (6.00%)	15 (9.50%)	ns
ATM	7 (3.24%)	7 (4.43%)	ns
POLE	4 (1.85%)	1 (0.63%)	ns
BRCA1	4 (1.85%)	1 (0.63%)	ns
BRCA2	6 (2.80%)	2 (1.30%)	ns
BARD1	2 (0.93%)	5 (3.20%)	ns
BRIP1	3 (1.40%)	3 (1.90%)	ns

*** indicates *p* < 0.001. ns indicates not significant.

**Table 3 cancers-14-04282-t003:** Clinicopathological features and risk factors of 3 subgroups in the low-repair group.

	L1	L2	L3	*p*-Value
*n*	86	56	74	
Age				
Mean (± SD)	60.30 (± 15.04)	58.23 (± 13.84)	63.15 (± 12.23)	0.126
Gender				
Female	24 (36.9)	18 (27.7)	23 (35.4)	0.843
Male	62 (41.0)	38 (25.2)	51 (33.8)	
Race ^a^				
White	56 (47.9)	21 (17.9)	40 (34.2)	0.002 *
Asian	21 (26.9)	29 (37.2)	28 (35.9)	
Black	5 (50.0)	0 (0.0)	5 (50.0)	
Molecular subtype				
HCC	81 (40.9)	50 (25.3)	67 (33.8)	0.192
Blast-Like	1 (11.1)	5 (55.6)	3 (33.3)	
CCA-Like	4 (44.4)	1 (11.2)	4 (44.4)	
AJCC Pathologic Tumor Stage ^b^				
I	45 (38.8)	27 (23.3)	44 (37.9)	0.446
II	17 (38.6)	16 (36.4)	11 (25.0)	
III/IV	16 (39.0)	12 (29.3)	13 (31.7)	
AJCC Pathologic Primary Tumor (pT) ^c^				
T1	48 (64.0)	26 (54.2)	46 (64.8)	0.691
T2	15 (20.0)	13 (27.1)	12 (16.9)	
T3/T4	12 (16.0)	9 (18.8)	13 (18.3)	
Grade ^d^				
G1/G2	74 (47.7)	33 (21.3)	48 (31.0)	<0.001
G3/G4	10 (17.5)	23 (40.4)	24 (42.1)	
Cirrhosis/Fibrosis ^e^				
Cirrhosis/Fibrosis	31 (36.5)	20 (23.5)	34 (40.0)	0.028
No	27 (55.1)	13 (26.5)	9 (18.4)	
HBV status ^f^				
Negative	69 (47.3)	30 (20.5)	47 (32.2)	0.001
Positive	15 (22.7)	26 (39.4)	25 (37.9)	
Vascular invasion ^g^				
None	21 (29.6)	18 (39.1)	23 (38.3)	0.46
Micro/Macro	50 (70.4)	28 (60.9)	37 (61.7)	

^a^ Excludes samples noted as other and NA (*n* = 9). ^b^ Excludes samples without pathologic stage annotation (*n* = 15). ^c^ Excludes samples based on AJCC 4th and 5th editions and noted as NA (*n* = 22). ^d^ Excludes samples without grade annotation (*n* = 4). ^e^ Excludes samples without annotation (*n* = 82). ^f^ HBV infection determined by >5 HBV reads from RNA-seq. ^g^ Excludes samples without annotation (*n* = 39). * *p*-value based on Asian vs. white people.

## Data Availability

The TCGA data are available through the GDC data portal, https://portal.gdc.cancer.gov/, Last Accessed Date 2 June 2022.

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
