# Peer review of "DNA Damage Repair Classifier Defines Distinct Groups in Hepatocellular Carcinoma"

_cancers, 2022, doi:10.3390/cancers14174282_

Round 1

Reviewer 1 Report

The work by Smith et al. was focused on the study of DNA damage repair pathways in hepatocellular carcinoma (HCC). By using the TCGA HCC cohort, which includes the RNAseq data from 374 patients with HCC, the authors evaluated the expression of a curated panel of 199 genes related to 15 DNA repair pathways as a way to identify novel potential subclasses and associations with clinicopathological findings. In this elegant study, the authors describe 2 distinct transcriptional subclasses: high repair and low repair. They observed that the “high repair” subclass was associated with more clinically-aggressive tumor features and prognosis, TP53 mutations, the Asian ethnicity and HBV infections. On the other hand, the “low repair” group was further subclassified in 3 distinct subgroups (L1, L2 and L3) from which L3 was show to be an intermediary group between and low repair groups. Overall, the work is extremely well-written and very-well designed and the results are novel and sound. I only have some minor points to be addressed before publication. 

-       Regarding the 199 DNA repair genes, how did the authors select them? Please provide the full list. 

-       The authors only made associations with HBV infections. What about other etiologies? Are there any associations with HCV, NAFLD, alcohol intake, etc?

-       Although the information is not available in the TCGA cohort, an important feature of chemoresistance in tumors is enhanced DNA repair. The authors should discuss if the high repair tumors are more chemoresistant. Are there any differences in transporter pumps and other resistance-related genes between groups?

-       The authors should provide in discussion a possible translation of their findings into clinics. How this new stratification may help in clinical practice? Do the authors may suggest any particular genes/proteins to be included in the measurements in clinical practice?

Reviewer 2 Report

Authors in this study provided important information about distinct groups of HCC which is based on the DNA repair pathways. I have minor comments which are given below:

1. Can the author's comment on the tumor immune microenvironment of these distinct subgroups identified based on DNA repair pathways. 

2. Can the information provided in this study can be applied in the clinics? Can it be used to predict the patient response to a particular drug which targets DNA repair pathways in-particular if the patient falls in the category of high DNA damage response group.
